# A Systematic Review of Oral Vertical Dyskinesia (“Rabbit” Syndrome)

**DOI:** 10.3390/medicina60081347

**Published:** 2024-08-19

**Authors:** Jamir Pitton Rissardo, Krish Kherajani, Nilofar Murtaza Vora, Venkatesh Yatakarla, Ana Letícia Fornari Caprara, Jeffrey Ratliff, Stanley N. Caroff

**Affiliations:** 1Neurology Department, Cooper University Hospital, Camden, NJ 08103, USA; ana.leticia.fornari@gmail.com; 2Medicine Department, Terna Speciality Hospital, Navi Mumbai 400706, India; krish.kherajani@gmail.com (K.K.); nilofar031202@gmail.com (N.M.V.); yatakarlavenkatesh7@gmail.com (V.Y.); 3Neurology Department, Thomas Jefferson University, Philadelphia, PA 19107, USA; jeffrey.ratliff@jefferson.edu; 4Psychiatric Department, University of Pennsylvania Perelman School of Medicine, Philadelphia, PA 19104, USA; caroffs@pennmedicine.upenn.edu

**Keywords:** rabbit syndrome, parkinsonism, dyskinesia, tardive dyskinesia, drug-induced, antipsychotic, antidepressant

## Abstract

*Background and Objectives:* Vertical rhythmic dyskinetic movements that are primarily drug-induced and affect solely the jaw, mouth, and lips without involving the tongue have been historically described as “rabbit” syndrome (RS). Evidence on the unique features and implications of this disorder remains limited. This literature review aims to evaluate the clinical–epidemiological profile, pathological mechanisms, and management of this movement disorder. *Materials and Methods:* Two reviewers identified and assessed relevant reports in six databases without language restriction published between 1972 and 2024. *Results:* A total of 85 articles containing 146 cases of RS were found. The mean frequency of RS among adults in psychiatric hospitals was 1.2% (range 0–4.4%). The mean age of affected patients was 49.2 (SD: 17.5), and 63.6% were females. Schizophrenia was the most frequent comorbidity found in 47.6%, followed by bipolar disorder (17.8%), major depressive disorder (10.3%), and obsessive–compulsive disorder (3.7%). Five cases were idiopathic. The most common medications associated with RS were haloperidol (17%), risperidone (14%), aripiprazole (7%), trifluoperazine (5%), and sulpiride (5%). The mean duration of pharmacotherapy before RS was 21.4 weeks (SD: 20.6). RS occurred in association with drug-induced parkinsonism (DIP) in 27.4% and with tardive dyskinesia (TD) in 8.2% of cases. Antipsychotic modification and/or anticholinergic drugs resulted in full or partial recovery in nearly all reported cases in which they were prescribed. *Conclusions:* RS occurs as a distinct drug-induced syndrome associated primarily but not exclusively with antipsychotics. Distinguishing RS from TD is important because the treatment options for the two disorders are quite different. By contrast, RS may be part of a spectrum of symptoms of DIP with similar course, treatment outcomes, and pathophysiology.

## 1. Introduction

In 1972, “rabbit” syndrome (RS) was first coined by Villeneuve et al. [1]. It is characterized by vertical rhythmic dyskinetic movements, affecting solely the jaw, mouth, and lips without involving the tongue. This involuntary motor disorder is defined by swift and precise movements of the muscles in the mouth and jaw along its vertical axis, resembling the chewing motions of a domestic rabbit occurring approximately five times per second [2]. Also, RS is often associated with popping sounds produced by the lips.

Factors such as fatigue, anxiety, attention, motor skills, and stressful situations can exacerbate RS symptoms by increasing muscle tone [3]. RS is often misdiagnosed as orofacial tardive dyskinesia (TD). However, classic TD is usually characterized by slower and less rhythmic movements of the mouth, lips, and jaw [4]. The symptoms of RS may arise because of antipsychotic and antidepressant use [5].

RS is considered rare, highly specific concerning the affected muscle groups, and selective regarding its pharmacological relation to the cholinergic system. This literature review describes the clinical manifestations and frequency of RS while focusing on the pharmacology of the offending medications. We comment on the literature of clinical cases and discuss current management options for RS.

## 2. Materials and Methods

### 2.1. Search Strategy

We searched six databases to locate all the existing reports on RS published from 1972 to June 2024 in electronic form. Excerpta Medica (Embase), Google Scholar, Latin American and Caribbean Health Sciences Literature (Lilacs), Medline, Scientific Electronic Library Online (Scielo), and Science Direct were searched. The search term was rabbit syndrome (Appendix A). The Preferred Reporting Items for Systematic Reviews and Meta-Analyses (PRISMA) 2020 checklist was followed [6].

### 2.2. Inclusion And Exclusion Criteria

The inclusion criteria covered case reports, case series, original articles, letters to the editor, bulletins, and poster presentations, published from 1972 to June 2024, without language restriction to ensure a thorough review. When the non-English literature was beyond the authors’ proficiency (English, French, and Spanish) or the English abstract did not provide enough data, such as articles in Japanese and Korean, the Google Translate service was used [7].

The authors independently screened the titles and abstracts of all articles from the initial search. Disagreements among authors were solved through discussion. Cases not accessible by electronic methods, including after a formal request to the authors, were excluded. Cases with more than one factor contributing to RS were evaluated based on the probability of the event occurrence using the Naranjo algorithm.

### 2.3. Data Extraction

Data abstraction was performed. When provided, we extracted author, year of publication, patient’s demographic characteristics (age and sex), main diagnosis for prescription of medication, cause of RS, duration of pharmacotherapy, management after diagnosis of RS, outcome (no recovery, partial recovery, full recovery), parkinsonian symptoms (resting tremor, bradykinesia, rigidity), and dyskinetic symptoms (dyskinetic movements in extremities, trunk, face, and tongue). The data were extracted by two independent authors and double-checked to ensure matching.

A total of 6334 articles were found; 6001 were inappropriate, and 248 were unrelated to the subject, duplicates, inaccessible electronically, or provided insufficient data (Figure 1). A total of 85 articles containing 146 cases of RS were published in the literature and met the inclusion criteria.

### 2.4. Statistical Analysis

Categorical variables were represented as proportions. Continuous variables were represented as means, standard deviation (SD), median, and range. Statistical analysis was performed using Microsoft Excel Spreadsheet Software version 16.0 (Microsoft Corp, Redmond, WA, USA). No statistical comparisons among groups were performed as this was a descriptive study of uncontrolled published clinical case reports. For the variable regarding the duration of pharmacotherapy before the RS onset, a statistical model was designed to control outliers to obtain the mean and SD. Also, the mean and standard deviation of the frequency of RS were adjusted according to the sample sizes of the studies from the literature.

### 2.5. Definitions

The clinical characteristics and definitions of the RS were obtained from Villeneuve et al. [1]. The clinical diagnosis for the psychiatric conditions was obtained from the *Diagnostic and Statistical Manual of Mental Disorders* (Fifth Edition) [8]. The Naranjo algorithm was used to determine the likelihood of whether an adverse drug reaction was actually due to the drug rather than the result of other factors [9].

Unfortunately, the term “rabbit” syndrome was described historically and has become conventional in the literature. Patients with RS and caregivers might find this term inappropriate. Here, we decided to continue to use “rabbit” syndrome or RS to be consistent with past literature but chose to mention “rabbit” in quotes to start and subsequently refer to it as “RS”. We also propose the alternative term, oral vertical dyskinesia (OVD), as a more neutral descriptive term, for future studies.

## 3. Results

### 3.1. Frequency of RS

RS is believed to be uncommon, affecting only a small segment of psychiatric patients using antipsychotic medications. In the first prevalence studies, Yassa et al. examined the inpatient population of a psychiatric facility [10]. They discovered a 2.3% frequency of RS among 266 inpatients receiving older antipsychotics, either alone or combined with anticholinergic agents [10]. Among those exclusively on antipsychotics, the prevalence rose to 4.4% [10]. Another study found a 1.5% incidence of RS among geriatric individuals in the use of antipsychotics in a mental health clinic [11]. To date, no prevalence studies have been published on the correlation between RS and atypical antipsychotics (Table 1). Our analysis revealed a frequency varying from 0 to 4.4%, with a mean of 1.2%, in the adult population involving patients from psychiatric hospitals. The studies included were cross-sectional studies performed in a similar setting but in different ethnicities. Noteworthy, prevalence estimates from the 1980s/1990s may not generalize to the current era when second-generation antipsychotics are used. Furthermore, the prevalence estimates for RS among patients treated with first-generation antipsychotics from the 1980s/1990s may not even generalize to the present, even for patients treated with first-generation antipsychotics today, because prescribing practices have shifted (generally, less aggressive dosing overall with first-generation antipsychotics compared to the 1980s).

Even though RS is assumed to be a rare disorder, some factors could lead to misleading reporting data. First, there is a significant number of individuals with neuropsychiatric disorders who are started simultaneously on antipsychotics and anticholinergic medications before they develop any motor symptoms, possibly preventing or masking the appearance of RS. Second, RS may not be a well-known adverse event that is often overlooked and mistakenly confused with TD. Thus, some individuals misdiagnosed with TD may have RS instead. Third, the syndrome can have mild symptoms, being elicited only by distracting tasks. Fourth, most scales that assess extrapyramidal symptoms do not include a specific item about RS. Interestingly, a rating scale for TD proposed by Simpson et al. in 1979 includes the description of “tremors of the upper lip (rabbit syndrome)”, classifying the severity on six levels and with an interrater reliability of 84% [14]. Failure to recognize RS and overdiagnosis of TD can lead to withholding highly effective treatment with anticholinergic agents, unnecessarily reducing the dose of required antipsychotics, or even prescribing vesicular monoamine transporter 2 (VMAT2) inhibitors with adverse consequences.

### 3.2. Patient Risk Factors

Some of the general risk factors associated with drug-induced movement disorders are advanced age, female sex, affective disorder diagnosis, greater total drug exposure, and prior medical history of drug-induced parkinsonism (DIP) [15]. In this context, the risk factors associated with RS are similar to drug-induced movement disorders. Thus, RS predominantly occurs in middle-aged and elderly individuals [16], with women being at a higher risk compared with men [17]. In this review, the mean and median age were 49.2 (SD: 17.5) and 55 years (range: 4 years–90 years). Sex was reported in 121 cases, of which 77 were female (63.6%).

RS is most frequently reported in patients receiving antipsychotic treatment for schizophrenia, the most common indication for antipsychotic treatment often at high doses. However, it has also been described in patients with bipolar affective disorders, Korsakoff’s syndrome, and cognitive impairment who were taking antipsychotics. Therefore, RS is associated with antipsychotic treatment, regardless of the underlying cause for such treatment [18]. Additionally, previous brain injuries have been associated with RS [1].

The comorbidity most likely associated with the indication of therapy and etiology of RS was reported in 107 cases. Schizophrenia was the most frequently reported comorbidity found in 47.6% (51 cases), followed by bipolar disorder (19 cases), major depressive disorder (11 cases), obsessive–compulsive disorder (4 cases), cognitive impairment (3 cases), acute psychotic disorders (3 cases), suicide attempt (2 cases), aggressive behavior (2 cases), hair loss (2 cases), headache (2 cases), substance use disorder (1 case), attention-deficit/ hyperactivity disorder (1 case), chronic hepatitis C (1 case), and gastrointestinal disorder (1 case). Five cases were considered idiopathic.

Asian individuals have been reported to be more vulnerable to DIP than Black or White patients [19]. Interestingly, the majority of the RS reports are from Asian countries. Therefore, some authors believe that Asian ethnicity is a risk factor for RS [20]. Also, the fact that some individuals developed abnormal movements with different medications before RS shows the importance of possible genetic factors [21].

### 3.3. Treatment Risk Factors

Antipsychotics were the most common class of medications associated with RS. However, this syndrome has been reported with other classes of medication, and even idiopathic cases were described (Appendix A [2,10,11,12,13,16,17,20,21,22,23,24,25,26,27,28,29,30,31,32,33,34,35,36,37,38,39,40,41,42,43,44,45,46,47,48,49,50,51,52,53,54,55,56,57,58,59,60,61,62,63,64,65,66,67,68,69,70,71,72,73,74,75,76,77,78,79,80,81,82,83,84,85,86,87,88,89,90,91,92,93,94,95,96,97]). The most common antipsychotics associated with the occurrence of RS were atypical antipsychotics in 42 cases, of which 15 cases were related to risperidone. Also, 34 individuals were taking typical antipsychotics, of which 18 were associated with haloperidol. The description of antidepressants was related to 11 cases (Table 2). Moreover, the antipsychotic dose, especially for risperidone, was related to a higher frequency of cases of RS. Paliperidone is the primary active metabolite of risperidone and has also been associated with RS [22]. Another specific risk factor for RS secondary to risperidone is the poor metabolizers of the cytochrome P450 2D6 isoenzyme, which can increase risperidone levels and, consequently, its adverse reactions [23].

#### 3.3.1. Typical Antipsychotics

In 1992, Schwartz et al. found 34 cases reported in the literature, of which 12 were associated with haloperidol and 8 with piperazine phenothiazines (fluphenazine, perphenazine, and trifluoperazine) [98]. High-potency antipsychotics are recognized for their tendency to cause DIP [98]. The fact that some patients who developed RS had previously experienced typical secondary parkinsonian side effects [33] may support the assumption that RS and DIP might share a common mechanism mediated by dopaminergic neuron blockage in the extrapyramidal system [25]. Most of the cases involving antipsychotics are related to high-potency dopaminergic blockage and low anticholinergic activity, such as haloperidol. Goswami et al. reported a prospective study with 19 cases of RS identified during 13 years [36]. All the patients used typical antipsychotics. RS did not persist during sleep stages II, III, and IV [36]. Also, the authors observed that all the patients received intravenous promethazine with full symptomatic recovery [36].

#### 3.3.2. Atypical Antipsychotics

Atypical antipsychotics are characterized by a comparatively lower frequency of movement disorders [99]. However, in our study, we found 15 patients with RS secondary to risperidone. The high incidence of RS associated with risperidone might be related to its strong affinity for D2 receptors, coupled with its low affinity for cholinergic muscarinic receptors. Interestingly, risperidone affects 5-HT2A receptors linked to antipsychotic action and relief of some of the movement disorders experienced with typical antipsychotics [100,101].

Paliperidone, an active metabolite of risperidone, has a slightly different pharmacodynamic profile from risperidone. For example, paliperidone has a more potent dopamine D2 receptor occupancy and weaker serotonin 5-HT2A receptor occupancy than risperidone. Interestingly, Teng et al. reported a case of RS occurring with paliperidone but not with risperidone, which highlights that the incidence of antidopaminergic adverse effects could still differ between risperidone and paliperidone in susceptible patients [69].

Kannarkat et al. assessed recent antipsychotics and their risk for the development of drug-induced movement disorder [102]. They found that lumateperone and pimavanserin show promise in being able to treat psychosis while reducing the risk of drug-induced movement disorder [102]. Long-acting paliperidone may reduce the risk of drug-induced movement disorder [102]. In contrast, other long-acting injectable formulations of second-generation antipsychotics have a similar risk of drug-induced movement disorder compared to oral formulations [102].

#### 3.3.3. Antidepressants

The literature on RS is scarce regarding other classes of medications besides antipsychotics. Epidemiologic studies suggest that abnormal involuntary movements occur in about 1 in 1000 adult patients treated with selective serotonin reuptake inhibitors [103]. Also, the incidences of akathisia, dystonia, and parkinsonism associated with selective serotonin reuptake inhibitors (SSRIs) are 45%, 28%, and 14%, respectively [104]. A few clinical cases of RS were triggered by SSRIs like citalopram, escitalopram, or paroxetine [54,59]. Earlier, cases of RS symptoms had already been associated with tricyclic antidepressants (TCAs). Also, a dose-dependent effect was observed with SSRIs [94].

Sastry et al. reported a case of RS secondary to escitalopram, which was given to a patient with vertigo on previous use of flunarizine [92]. The patient developed a significant hypersensitivity reaction to escitalopram [92]. The authors reported that sleep suppressed RS symptoms, but no differentiation among the sleep stages was described [92].

#### 3.3.4. Psychostimulants

There is a report of RS secondary to methylphenidate [60]. Notably, most cases of orofacial movements associated with methylphenidate are oromandibular dystonia and not dyskinetic movements [105]. Therefore, a detailed description of the phenomenology and the inclusion of electrodiagnostic studies can help distinguish these two pathologies.

#### 3.3.5. Idiopathic Cases

Nishiyama et al. reported a case of RS considered idiopathic [35]. Noteworthy, brain magnetic resonance imaging revealed atrophy of the cerebellum and brainstem [35]. An interesting fact was the varying effects of drugs on the clinical manifestations presented by affected individuals [35]. For example, while atropine and trihexyphenidyl did not affect RS symptoms, levodopa worsened the condition [35]. Haloperidol, sulpiride, and chlorpromazine were effective in decreasing the frequency of vertical lip movement [35]. Miwa et al. reported another idiopathic case where diazepam, trihexyphenidyl, levodopa, and β-adrenoceptor antagonists did not improve RS symptoms [44]. Also, Truong et al. described an idiopathic case of a patient with a prior medical history of meningioma who significantly improved after botulinum toxin therapy [32]. Differences in pharmacological responses between idiopathic cases and drug-induced RS might suggest different underlying mechanisms.

## 4. Pathophysiology

The exact mechanism of RS remains unknown, and the literature suggests conflicting causes. It has been suggested that RS may be similar to or a forme fruste of DIP, whereby a hypercholinergic state arises secondary to dopamine blockade [17]. Alternatively, it has been postulated that the mechanism is similar to that of TD, characterized by a state of cholinergic hypofunction due to dopaminergic hypersensitivity. Increasing confusion, and similar to TD, RS can manifest during antipsychotic treatment or after discontinuation of such medications. Therefore, the etiopathogenesis of RS, DIP, and TD remains unclear.

### 4.1. Cases Related to Relative Hypercholinergic and Hypodopaminergic States

The hypercholinergic state induced by dopaminergic pathway blockage in the extrapyramidal system may explain the responsiveness of RS to anticholinergic drugs [25,33]. Interestingly, there is a report of RS associated with interferon [83]. This case can also be explained by the dopaminergic pathway since interferon-alfa administration was observed to inhibit dopaminergic activity, decreasing dopamine levels in mice brains [106].

Nishiyama et al. reported a case of RS successfully treated with haloperidol in a woman with multiple system atrophy [35]. In this case, the etiopathogenetic mechanism might resemble that of TD, possibly indicating a cholinergic hypofunction due to denervation-type dopaminergic hypersensitivity in the basal ganglia [107].

First-generation antipsychotics are dopamine receptor antagonists and block histamine, muscarinic, and α-1 receptors [108], whereas second-generation antipsychotics are serotonin–dopamine antagonists [109]. Serotonin antagonism might enhance dopaminergic neurotransmission in the nigrostriatal pathway, thereby reducing the risk of extrapyramidal side effects such as dystonic reactions, akathisia, and TD [109]. TCAs inhibit serotonin and norepinephrine reuptake by presynaptic neurons, leading to increased levels of these neurotransmitters in the synaptic cleft. TCAs also exhibit a high affinity to muscarinic, cholinergic, and adrenergic receptors, while SSRIs inhibit serotonin reuptake in the synaptic gap. Anticholinergics such as trihexyphenidyl and procyclidine may manage the hypercholinergic state in RS [31].

Kapur et al. found that the likelihood of abnormal movements increased significantly as D2 occupancy exceeded 78%, even with atypical antipsychotics [110]. Interestingly, among atypical antipsychotics, risperidone has a D2 occupancy of 75%, a predicted D3 occupancy of 38%, and a D2:D3 selectivity of 4.9 [111]. So, these facts may explain the high number of cases of RS secondary to risperidone if RS was presumably related to a direct dopaminergic pathway.

### 4.2. Cases Unrelated to Direct Hypercholinergic and Hypodopaminergic States

Interestingly, antipsychotic exposure was found to be unrelated to some documented cases of RS. Truong et al. described a case of a 76-year-old female who had undergone brain surgery 16 years earlier [32]. Kamijo et al. reported a case related to the ingestion of a large quantity of phenol in a suicide attempt, possibly inducing RS by a hypercholinergic pathway [42]. RS was also observed in patients treated with imipramine, citalopram, paroxetine, and methylphenidate [16,59,60]. It is conceivable that these antidepressants, potent inhibitors of serotonin reuptake, may induce RS through serotonin-mediated inhibition of dopaminergic neurotransmission in predisposed individuals [112,113]. However, some patients showed improvement with prescription SSRIs, such as fluvoxamine [85]. A possible explanation for improving RS with fluvoxamine is the potent σ-1 chaperone receptor agonistic action [94]. σ receptor agonism has various effects via calcium signaling, protein kinase translocation or activation, ion channel firing, neurotransmitter release, cellular differentiation, neuronal survival, and synaptogenesis [114].

Datta et al. reported an interesting case of anti-N-methyl-D-aspartate receptor (NMDAR) antibody encephalitis that developed into RS [88]. It is well-known that anti-NMDAR encephalitis can have perioral manifestations [88]. One possible explanation for this case is seizure activity, but the authors did not perform electrodiagnostic studies to explain the pathophysiology fully [88].

Park et al. described an individual developing RS secondary to medullary compression by the vertebral artery [80]. The authors believed the pathophysiology could be similar to those triggering palatal tremor [80]. Medullary compression may mediate the interruption of the dentato-rubro-olivary pathway by a gross vascular indentation on the Gullain–Mollaret triangle [80]. It is worth mentioning that the palatal tremor is characterized by a rhythmic 0.5–3 Hz, which is quite lower than that observed in RS [115].

There is a significant overlap between RS and functional disorders. Thus, case reports of idiopathic RS should provide a detailed description involving electrodiagnostic and neuroimaging studies. Nagar et al. reported two cases of high doses of minoxidil foam leading to RS [78]. However, the authors did not provide information regarding the phenomenology of the movement or neurologic studies. Also, there is no report in the literature on minoxidil’s effect on dopaminergic levels.

SSRIs are a class of medication frequently found in the literature associated with RS. Depending upon experimental conditions, 5-hydroxytryptophan (5-HTP) and serotonin can create gnawing movements in rats, but these are different from those provoked by dopamine [116]. In animals, apomorphine and amphetamine can also trigger these movements in the jaw [117].

### 4.3. Rabbit Syndrome, Drug-Induced Parkinsonism, and Tardive Dyskinesia

It is not known whether RS poses a risk for the later development of TD. In this context, Schwartz et al. reported a middle-aged female being treated with haloperidol who developed RS in the absence of other extrapyramidal symptoms. Trihexyphenidyl was started with full recovery of RS symptoms [38]. Nevertheless, one month later, the patient developed TD [38]. Schwartz et al. associated the “uncovering” of TD with the anticholinergic treatment; however, the temporal association between RS and the subsequent development of TD was not considered [38]. In our review, 12/146 (8.2%) of patients with RS were reported to have concurrent signs of TD (Appendix A).

Kachi et al. suggested that RS is a transitional form from DIP to TD because the intravenous administration of haloperidol has shown a suppressive effect on RS [28]. Also, Wada et al. reported patients with RS and parkinsonian symptoms simultaneously, and the improvement in RS occurred in association with the improvement in the accompanying parkinsonian symptoms [17]. These findings provide further evidence that the pathophysiology of RS may have similarities to that of acute forms of DIP [17]. In our review, 40/146 (27.4%) patients had concurrent parkinsonian signs (Appendix A).

## 5. Clinical Features

RS is defined by swift and precise movements of the muscles in the mouth and jaw along its vertical axis, 3–5.5 Hz in frequency, resembling the chewing motions of a domestic rabbit [2]. Sometimes, squinting of the nose with activation of the levator labii superioris and levator labii superioris alaeque nasi muscles can also be noticed [79]. However, it is worth mentioning that a uniform and accepted phenomenological description of the syndrome is still needed.

Some authors proposed other lip movements besides the vertical axis, but these would be better characterized as dyskinesias. For example, there is a description of isolated horizontal movements of the lips [118] and a report of only upper lip movement [92]. The rating scale for TD proposed by Simpson et al. in 1979 describes RS as “tremors of the upper lip [14]”. One interesting clinical feature of RS is that the frequency of the lip movements is similar to the tremor found in patients with Parkinson’s disease, whose dominant frequency is 3.5–7.5 Hz [119]. Therefore, some authors believe in the idea of a spectrum involving Parkinson’s disease, RS, and TD.

Interestingly, one patient illustrated the symptoms of RS as “sending a kiss to someone [62]”. Other individuals described it as “trembling movements” of the lips [16] and “vibration of my lips [33]”. Also, Parvin et al. reported that a patient described her experience of the symptoms as “I have dry mouth [59]”.

The literature is contradictory regarding the effect of speech on RS. Gangadhar et al. reported that RS symptoms were rhythmic and occurred more at rest, which tended to diminish while engaged in conversation [27]. However, some authors reported these abnormal movements became even more accentuated and rapid when the patient attempted to talk [1]. Also, Villeneuve et al. described that only the muscles of the inferior part of the face were involved, particularly the superficial muscles of the mouth and the masseters. No lateral displacement of the inferior mandible was noticed [1].

The vertical rhythmic movement is often associated with popping sounds produced by the lips, which appear, in particular, while speaking or under duress. Some patients describe these sounds as “strange and irritating [43]”.

Villeneuve et al. evaluated patients with RS, asking them to navigate a labyrinth pattern with a pencil while performing the palmo-mental and labio-oral reflexes. These maneuvers usually increased the frequency of lip movements [1]. Also, Villeneuve et al. first hypothesized that the presence of palmo-mental and labio-oral reflexes was associated with central nervous system lesions [1]. Currently, it is well known that this assumption is not always true and that these reflexes have nonspecific neuroanatomic correlation.

Similar to TD and movement disorders in general, RS movements become more pronounced in situations involving fatigue and anxiety. Stressful circumstances can act as triggers, exacerbating the symptoms of RS. RS commonly worsens during tasks requiring focused attention, concentration, and motor skills, often accompanied by increased muscle tone, and conversely diminishes with rest, calmness, and sleep [3].

Wada et al. proposed a classification of the severity of RS according to vertical lip movements and finger tapping [17]. RS was rated on a scale from 0 to 3 (Table 3). Other scales that were commonly used to define the severity of RS were the abnormal involuntary movement scale (AIMS) and the Simpson–Angus Scale (SAS) [17].

Most patients with RS also show some parkinsonian signs, such as lead-pipe rigidity, bradykinesia, and limb tremor (27.4% in our review). Also, RS symptoms share similarities with Parkinson’s disease symptoms in that they persist during stage 1 non-rapid eye movement (NREM) sleep [120]. However, Gangadhar et al. reported that RS was not constantly present in stage I NREM sleep [27]. There were occasions when movement was completely absent during this stage of sleep, and this disappearance was patchy with no predilection for early, middle, or late parts of this stage of sleep [27]. It is worth mentioning that TD symptoms typically cease during stage I NREM sleep [121].

Fornazzari et al. reported a single-photon emission computed tomography (SPECT) scan utilizing technetium-99m (99mTc) hexamethylpropyleneamine oxime (HMPAO) upon initial assessment and at the 6-month follow-up [16]. RS revealed decreased basal ganglia perfusion when lip movements were presented [16]. On the other hand, a return to normal perfusion when RS resolved was observed [16]. Kuo et al. reported a case of RS likely associated with stroke [84]. A 99mTc ethyl cysteinate dimer SPECT revealed increased perfusion of the right basal ganglia and left thalamus [84]. Levodopa/ benserazide was started with full improvement in RS. After approximately one month, a repeated SPECT showed that the basal ganglia perfusion had returned to normal [84]. Noteworthy, increased basal ganglia perfusion has been reported in patients with schizophrenia, particularly those with auditory hallucinations [122].

Gastrointestinal complications leading to weight loss were previously described with severe TD. Some patients with RS reported significant dysphagia. Hayashi et al. reported a case of RS with oxypertine, which improved after its discontinuation [40]. Interestingly, the case showed a polygraphic recording of before and after a biperiden injection of the thoracic muscles, and a significant reduction in the micro-vibrations from 3 to 0 Hz was clearly noted [40].

## 6. Management

Overall, 68 (46.6%) of the 146 reported cases were reported to have a full recovery, and 13 (8.9%) had partial improvement (Appendix A). When a model controlling outliers was applied, the mean duration of pharmacotherapy before RS was 21.4 weeks (SD: 20.6).

The first step in managing RS is identifying the offending agent and discontinuing or reducing its dose. However, the majority of the offending drugs are antipsychotics, and for patients with psychotic disorders, complete discontinuation of the antipsychotic is not feasible because of their underlying psychiatric conditions [73]. However, in patients without psychotic disorders who could be safely managed and for whom effective alternative treatments are available, reductions in doses, switches, or discontinuation of antipsychotics may be considered. Full or partial recovery was reported in 49 and 6 cases, respectively, after antipsychotic modifications (Appendix A).

The advent of atypical antipsychotics has prompted the adoption of alternative treatment approaches. Transitioning to a second-generation antipsychotic with strong anticholinergic properties, such as olanzapine or clozapine, might be another viable management strategy, considering their lower propensity to induce RS compared with risperidone [99]. Durst et al. reported a case of RS induced by the typical antipsychotic zuclopenthixol, which was managed by switching the antipsychotic medication to olanzapine [47]. Following olanzapine administration, RS and the concurrent psychotic symptoms exhibited significant improvement [47]. However, caution is warranted as rare cases have implicated olanzapine itself as a potential cause of RS [46].

Quetiapine has demonstrated efficacy as a monotherapy for managing both RS and psychotic symptoms. Altidang et al. reported a case of RS induced by risperidone and unresponsive to anticholinergic agents, resolved after the patient was transitioned to quetiapine. After 4 weeks of quetiapine therapy, RS movements markedly diminished, with continued improvement during subsequent follow-ups [56]. Additionally, the prescription of antiparkinsonian agents may be necessary [123], although RS has not always responded to levodopa or dopamine agonists [124].

After modification of the offending agent, an anticholinergic medication, such as procyclidine, trihexyphenidyl, or benztropine, can be prescribed [4]. Unlike TD, RS typically responds favorably to anticholinergic agents [25]. Trihexyphenidyl has been effectively utilized in severe cases progressing to dysphagia [40,48], for example, as well as in patients with schizophrenia receiving long-term antipsychotics, resulting in significant reductions in RS rating scale scores and Simpson–Angus rating scale [17]. RS symptoms usually improve within a few days after initiating anticholinergic treatment [38], although in some cases, the syndrome may reoccur upon discontinuation of anticholinergic medications [31]. Researchers have adjusted treatment strategies after observing that some RS patients treated with anticholinergic agents can develop TD later [26]. Full or partial recovery was reported in 41 and 10 cases, respectively, after anticholinergic treatment (Appendix A). In other words, although limited by publication bias of reporting only positive outcomes, 51 out of 53 (96.2%) patients in whom anticholinergic treatment was prescribed showed partial or full recovery. Therefore, the differentiation between RS and TD is important because there is a significant response to anticholinergic.

Although VMAT2 inhibitors have been approved as a treatment for TD, the response of RS to VMAT2 inhibitor therapy has not been formally tested. Because these agents deplete dopamine and have been associated with causing or worsening DIP, they would likely worsen RS, further emphasizing the need to distinguish RS from TD. In one reported case, tetrabenazine was prescribed in combination with promethazine and modification of olanzapine, resulting in full recovery of RS [89].

## 7. Differential Diagnosis

### 7.1. Oral Tardive Dyskinesia

The distinction between RS and TD is important because the treatment options for the two disorders are quite different (Table 4). The movements associated with RS are distinct from TD, the latter characterized by slower, less rhythmic, and consistent movements [31]. RS is characterized primarily by tremors of the chin, jaw, lips, and mouth. TD most often specifically affects the upper face, mouth, lips, jaw, and tongue, manifesting as stereotyped, choreiform, or dystonic involuntary movements. Chewing-type movements are described in TD as well but are slower, stereotyped, and not rhythmic. Stereotyped, dystonic, or choreoathetoic movements of other body parts are much more likely in TD than in RS, whereas RS is associated with other features of parkinsonism, such as bradykinesia and rigidity. Furthermore, anticholinergics can help in RS but worsen TD symptoms. Also, it is debatable whether persistent TD occurs with drugs other than dopamine blockers, whereas RS has been associated with antidepressants and stimulants.

RS can also be differentiated from typical oral dyskinesia because the latter is usually suppressible, and orofacial TD may occur in the tongue and is associated with dyskinesias in other facial areas and body parts [127].

It is worth mentioning that TD and RS may coexist in a patient and are not mutually exclusive. Twelve (8.2%) of the patients with RS in this review had concomitant signs of TD. Sovner et al. reported an interesting case of a patient who received benztropine, and her RS improved, but the dyskinesia in her extremities worsened [25]. On the other hand, Weiss et al. described a patient whose physostigmine was prescribed, and her RS worsened, but the dyskinesia in her extremities completely recovered [26].

Jus et al. assessed the effectiveness of benztropine, diazepam, diphenylhydantoin, and tryptophan on TD and RS [24]. They observed that benztropine was effective in treating RS but only produced short-term reductions in muscle tension in TD [24]. Diazepam was effective in both conditions but caused a simultaneous decrease in attention [24]. Diphenylhydantoin was effective in approximately half of the TD patients but ineffective in RS. Tryptophan was ineffective in both conditions [24].

### 7.2. Horizontal Perioral Movements

Singh et al. reported a case of an individual presenting with horizontal perioral movement, tongue involvement, and acute onset, which the authors defined as atypical RS [118]. They believe that this occurred because of the interaction between haloperidol and escitalopram [118]. Their case could be better classified as dyskinesia rather than atypical RS. However, constant horizontal movement could provide new insights into the differentiation of RS and TD. Also, it can explain the occurrence of atypical extrapyramidal features with a combined use of antipsychotics and antidepressants.

### 7.3. Bruxism

RS has clinical similarities with drug-induced bruxism, which is characterized by clenching of the jaw, tooth grinding, jaw pain, jaw spasms, facial pain, headaches, and sleep disorders. Drug-induced bruxism has been associated classically with anticonvulsants and antidepressants, particularly SSRIs such as fluoxetine and sertraline, though cases exist with dopamine blockers also [128]. The condition usually requires cessation of the culprit drug or the addition of other neurological drugs with partial agonistic action on 5-HT1a receptors, such as buspirone [129].

### 7.4. Oromandibular Dystonia

Oromandibular dystonia is a focal type of dystonia. Its phenomenology, characterized by contractions without a specific frequency, can be used to differentiate it from RS. Also, dystonic lateral jaw movement is not observed in individuals with RS. Noteworthy, both disorders will improve with the prescription of anticholinergic drugs [130].

### 7.5. Head Tremor in Parkinson’s Disease

The frequency of lip movements in RS is similar to that of tremors in Parkinson’s disease. In addition, patients with RS sometimes present with parkinsonian features, such as rigidity, bradykinesia, and hypomimia [131]. RS and DIP, which, together with idiopathic Parkinson’s disease, may represent different points along the pathophysiological continuum of the same disorder. Noteworthy, RS should be distinguished from isolated tremor syndrome. RS is the contraction of agonist muscles, and tremor is characterized by the activation of agonist and antagonist muscles entrained by a signal pattern originating from an oscillator in the central nervous system.

### 7.6. Functional Oral Movements

Buccal involuntary movements may occur in patients with schizophrenia before the onset of antipsychotic treatment, while RS is found almost always only after exposure to antipsychotics [124]. Aniello et al. reported an interesting cause of functional RS. Some maneuvers should be performed in the neurological examination of RS [90]. First, Aniello et al. observed the entrainment phenomenon, in which, when invited to perform simultaneous hand tasks at variable frequencies, movements of the mouth presented with the same variability as voluntary exercises performed by the hands [90]. Second, the movements of the lips decreased in frequency when performing complex patterns of finger opposition, known as distractibility [90]. Also, the authors observed inconsistency because the lip movements disappeared while the individual was talking [90].

## 8. The Missing Link

Instead of being a separate disorder, we propose that RS may be part of the spectrum involving DIP and TD (Figure 2). Three main factors can support this hypothesis. Currently, RS, DIP, and TD are considered different disorders, but based on our findings, several individuals had features of RS associated with DIP or TD, and some individuals developed the three disorders together.

First, there are reports of patients with RS developing parkinsonian features in the literature. Also, these cases had a more pronounced response using trihexyphenidyl. Moreover, the individuals only presented mild symptoms of parkinsonism. However, an attempt to categorize RS and DIP as variations of the same disorder has been deemed inappropriate by some authors since RS usually does not respond to treatment with dopamine agonists.

Second, some cases of RS with concurrent TD symptoms were reported. Cholinergic and anticholinergic medications may differentiate the two disorders, RS being responsive to anticholinergics, while there is suggested evidence that TD may respond to cholinergic agents. Furthermore, individuals with RS and TD were more likely to achieve partial recovery. Jus et al. revealed that patients with RS or TD do not have the same response to benztropine and diphenylhydantoin. Still, they can have the same response to tryptophan and diazepam [24].

Third, Villeneuve et al., Weiss et al., and Reichenberg et al. reported cases of patients presenting DIP, RS, and TD symptoms [1,26,82]. These reports are quite interesting because they can suggest a similar remarkable point for the pathophysiology of these disorders. Moreover, some authors reported RS as a risk factor for the later development of TD and DIP [28,38].

## 9. Future Studies

The terminology “rabbit” syndrome was unfortunately described in the literature and should be replaced by a more descriptive, neutral, and less pejorative term like oral vertical dyskinesia (OVD). Noteworthy, the disadvantages of such politically correct, but less descriptive altered terminology, make it harder to find relevant papers in literature databases for future scientists, as they would have to know all the different proposed possible designations of the disease in their literature search.

Moving forward, several critical areas warrant exploration to deepen our understanding and enhance management strategies for this syndrome. Firstly, delving into the genetic landscape of RS holds promise in unraveling its underlying mechanisms and identifying potential therapeutic targets. Genome-wide association studies and whole-genome sequencing efforts could uncover novel genetic variants associated with the disorder, paving the way for targeted interventions. No study in the literature specifically analyzed the genetic factors; only hypotheses were described.

Advanced neuroimaging techniques such as functional magnetic resonance and diffusion tensor imaging can elucidate the neural circuitry involved in RS. Noteworthy, there are only two studies in the literature with SPECT evaluating RS. These methods can provide insights into disease progression and help identify biomarkers for monitoring therapeutic responses. Also, conducting rigorous clinical trials to evaluate the efficacy and safety of pharmacological and non-pharmacological treatments is imperative for optimizing symptom management and improving the quality of life for individuals affected by RS. Several anecdotal reports have recommended treatment strategies for RS. An empirical investigation is needed to evaluate treatment strategies for RS. It is worth mentioning that clinical trials are likely unfeasible with RS because of low patient numbers and the large diversity throughout the cases.

Another important fact that should be studied is the effect of VMAT2 inhibitor therapy on RS. It is possible that VMAT2 inhibitors may worsen symptoms since these medications cause depletion of monoamines, reducing dopamine neurotransmission. Also, there are reports of parkinsonism associated with VMAT2 inhibitors. Therefore, the similarities between RS and DIP may presumptively imply that VAMT2 inhibitors may worsen symptoms of RS.

## 10. Conclusions

RS is considered a rare disorder with several factors that probably influence its epidemiology, including patient risk factors (age, sex) and treatment factors (dose and potency of antipsychotics). RS is most often reported as a distinct drug-induced movement disorder associated primarily with antipsychotics, but other drug classes and even idiopathic cases have been reported. We found that RS is frequently associated with both DIP and TD. RS appears to be part of the spectrum of DIP based on similar clinical features, acute onset and duration, and response to modifying antipsychotics or anticholinergic treatment. Like DIP, RS is likely predictive of the later onset of TD. Although there is a lesser risk of developing RS similar to other reversible drug-induced movement disorders, when using atypical antipsychotic medicines, clinicians should prescribe all such medications with caution. If detected early, RS is easily and effectively treated.

## Figures and Tables

**Figure 1 medicina-60-01347-f001:**
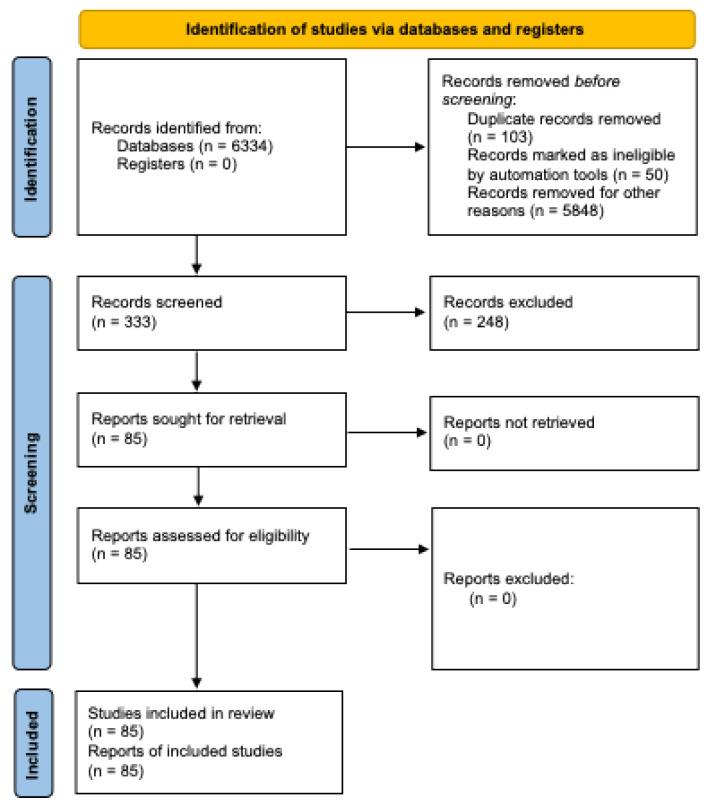
Flowchart of the screening process.

**Figure 2 medicina-60-01347-f002:**
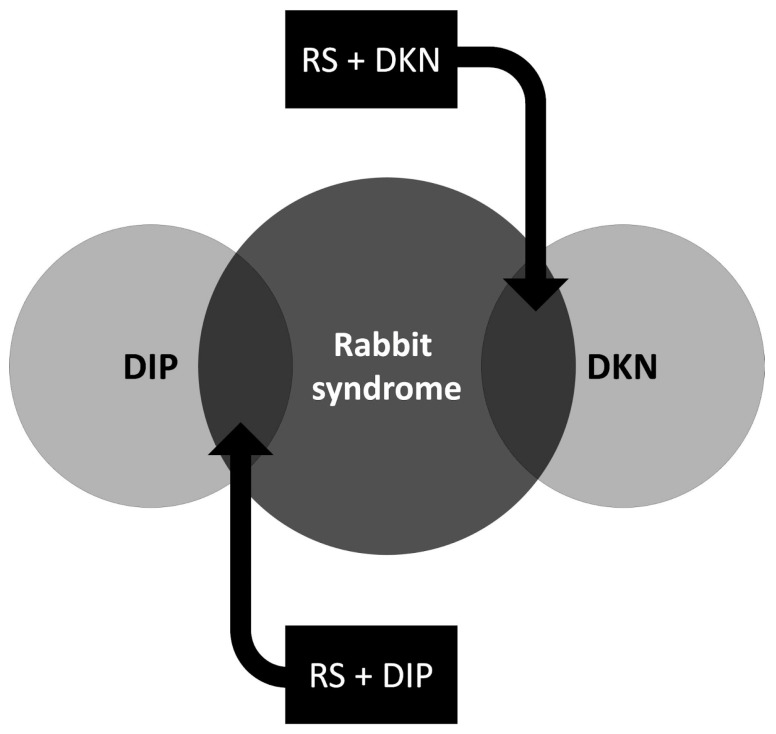
Drug-induced parkinsonism, rabbit syndrome, and dyskinesia spectrum. Abbreviations: DIP, drug-induced parkinsonism; DKN, dyskinesia; RS, rabbit syndrome.

**Table 1 medicina-60-01347-t001:** Frequency of RS among different studies.

Frequency	N ^I^	Total Sample ^II^	Note	References
4.4%	6	137	Patients from a psychiatric hospital taking only antipsychotics.	Yassa et al. (1986) [10]
2.4%	17	716	Patients diagnosed with several neuropsychiatric pathologies from a psychiatric hospital.	Inada et al. (1991) [12]
2.0%	3	149	Only patients diagnosed with schizophrenia from a psychiatric hospital.	Decina et al. (1990) [13]
1.5%	1	160	Patients diagnosed with several neuropsychiatric pathologies from a psychogeriatric clinic.	Chiu et al. (1993) [11]
0%	0	129	Patients from a psychiatric hospital taking antipsychotics and anticholinergics.	Yassa et al. (1986) [10]
0%	0	917	Patients diagnosed with several neuropsychiatric pathologies from a psychiatric hospital.	Chiu et al. (1993) [11]
0%	0	113	Healthy elderly individuals from a senior citizen center.	Chiu et al. (1993) [11]
1.2% ^III^	27	2321	Total samples	

^I^ Number of patients diagnosed with RS. ^II^ Total number of patients assessed in each study. ^III^ Mean of the frequency of all the samples combined. The mean and standard deviation (adjusted according to sample size) of the frequency were 1.163% and 1.161%.

**Table 2 medicina-60-01347-t002:** RS cases associated with medications.

Medication	Percentage	Number of Patients	Dose (mg/Day)
Haloperidol	17%	18	5–30 PO; 260 IM ^I^
Risperidone	14%	15	1–6
Aripiprazole	7%	7	10–20 PO; 400 IM ^I^
Trifluoperazine	5%	5	5–15
Sulpiride	5%	5	600–1200
Escitalopram	4%	4	10
Fluphenazine	4%	4	20 PO; 25–37.5 IM ^I^
Olanzapine	4%	4	10–20
Thioridazine	4%	4	25–100
Amisulpride	3%	3	150–800
Levomepromazine	3%	3	50–150
Quetiapine	3%	3	300–600
Amitriptyline	2%	2	32–75
Chlorpromazine	2%	2	50–150
Clozapine	2%	2	225–450
Minoxidil	2%	2	NA
Paliperidone	2%	2	3–4
Perphenazine	2%	2	2–8
Propericiazine	2%	2	50–60
Bromperidol	1%	1	20
Citalopram	1%	1	5
Clebopride	1%	1	1.5
Clocapramine	1%	1	100
Fluoxetine	1%	1	40
Flupenthixol	1%	1	NA
Imipramine	1%	1	150
Lithium	1%	1	600
Lofepramine	1%	1	210
Lurasidone	1%	1	120
Methylphenidate	1%	1	5
Mesoridazine	1%	1	100
Oxypertine	1%	1	60
Paroxetine	1%	1	20
Peginterferon alfa-2a	1%	1	0.18
Sertraline	1%	1	50
Zuclopenthixol	1%	1	200

Abbreviations: IM, intramuscular; NA, not available/not reported; PO, per os/per mouth. ^I^ Intramuscular haloperidol, aripiprazole, and fluphenazine are the long-acting injectable forms of the drug and, therefore, are given every few weeks.

**Table 3 medicina-60-01347-t003:** Rabbit syndrome rating scale by Wada et al. [17] (Wada, 1992) modified by Rissardo et al.

Score	Description
0	RS was absent even during finger-tapping.
1	RS was evident only during finger-tapping.
2	RS was present for less than half of the examination time.
3	RS was present for more than half of the examination time.

**Table 4 medicina-60-01347-t004:** Clinical features of RS and TD.

Disorder	Rabbit Syndrome	Tardive Dyskinesia	Reference
Movement frequency	Rapid, fine, and usually rhythmic	Slow, irregular (no rhythmicity)	Villeneuve et al. (1972) [1]
Movement axis	Vertical	All directions	Villeneuve et al. (1972) [1]
Tongue	No involvement	Usually, involved	Villeneuve et al. (1972) [1]
Associated abnormal movements	Bradykinesia, tremors, and rigidity	TD affects trunk, neck, and extremities	Villeneuve et al. (1972) [1]
Attention and voluntary movements, electromyography	Always increases in amplitude and potential frequency	Inconsistent findings	Jus et al. (1973) [3]
Transitory state from sleep to wakefulness	Persistent	Persistent	Jus et al. (1972) [121]
Stage I non-rapid eye-movement sleep	Usually persistent, but can be intermittent	Not persistent	Jus et al. (1973) [125] Gangadhar et al. (1981) [27]
Stages II, III, and IV non-rapid eye-movement sleep	Not persistent	Not persistent	Jus et al. (1972) [121]
Anticholinergic effect (trihexyphenidyl, benztropine)	Improves symptoms	Worsens symptoms	Sovner et al. (1977) [25]
Cholinergic effect (physostigmine)	Worsens symptoms	Improves symptoms	Weiss et al. (1980) [26]
Antipsychotic effect (haloperidol)	Worsens symptoms, but it can be improved	Suppresses symptoms	Gangadhar et al. (1981) [27]
Antipsychotic withdrawal	Improve symptoms	Initially worsens	Schwartz et al. (2004) [124]
Vesicular monoamine transporter 2 (VMAT2) effect (speculative)	Likely worsen	Likely improves	Stahl et al. (2018) [126]

## Data Availability

Not applicable.

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
