# Peer review of "A Systematic Review of Oral Vertical Dyskinesia (“Rabbit” Syndrome)"

_medicina, 2024, doi:10.3390/medicina60081347_

Round 1

Reviewer 1 Report

Comments and Suggestions for Authors

This is intererstin review according to Rabbit syndrome requency, risk factors and possible tretament - based on detailed review of described cases.

I have few comments - the main is -  is it narrative review or systemaic? if systematic it should be performed accoridng to PRISMA guidelines (with flow chart from PRISMA) as well as all systematic reviews should be with protocol registered previoulsy? 

The second could You clariify risk factors of RS - especially gender? and how to disinguish RS with tardie dyskinesia?

Author Response

REVIEWER 1

This is intererstin review according to Rabbit syndrome requency, risk factors and possible tretament - based on detailed review of described cases.

I have few comments - the main is -  is it narrative review or systemaic? if systematic it should be performed accoridng to PRISMA guidelines (with flow chart from PRISMA) as well as all systematic reviews should be with protocol registered previoulsy?

Authors: We review the review accordingly to the PRISMA guidelines, and we reference the article from 2020. We also included the checklist of the PRISMA features. We did not register this study, and by the PRISMA guidelines is only recommended and not mandatory.

The second could You clariify risk factors of RS - especially gender? and how to disinguish RS with tardie dyskinesia?

Authors: L152-153, we found “sex was reported in 121 cases, of which 77 were female (63.6%).” Also, the we provide a table with the features to distinguish between these two disorders (Table 4).

Reviewer 2 Report

Comments and Suggestions for Authors

In their manuscript, „A Literature Review Of Oral Vertical Dyskinesia (“Rabbit” 2 Syndrome) Rissardo et al. provide a systematic literature review on that highly rare disease.

       The last previous review on the topic is quite dated; thus the current manuscript is timely. It is also overall well written. The tables and the two figures are helpful also.

Major concern

1)    The pathophysiology and the possibly involved pathways with the many different cases mentioned are overwhelming and it is easy to lose sight of the overall picture. A Figure showing the various involved neurotransmitter pathways and the drugs interfering with those would thus be helpful (probably involving cortico-striatal, mesencephalic, basal ganglia pathways). Such a graph might possibly also be suitable as a graphical abstract of the manuscript.

Minor concerns

1)    It should be better explained how the criterion “insufficient data” e.g. in Fig. 1 as a reason for the exclusion of studies was defined.

2)    The quotation marks should be moved from line 60 to line 59

3)    Line 103-108 and 561-563 are a concession to the zeitgeist. It should also be mentioned that the disadvantages of such politically correct, but less descriptive altered terminology makes it harder to find relevant papers in literature databases for future scientists, as thus they would have to know all the different proposed possible designations of the disease in their literature search. Moreover, “rabbit syndrome” is much more intuitive than the proposed new wording even for the lay public.

4)    The authors mention several times that RS may be misdiagnosed as TD. The authors should check whether there are examples for such misdiagnosis in the literature, in addition to the 1979 paper by Simpson et al.

5)    The authors should also mention any data or experience with own patients how RS is being perceived by the patients, i.e. how bothersome it is usually perceived. For instance, such a description might fit in after lines 357-370. Such a description could be followed by an evaluation whether the side effects of e.g. anti-cholinergic treatment (e.g. on cognition), etc. are commensurate with the potential benefits.

6)    Line 258: Consider replacing “To increase the confusion” with “Increasing confusion,…”

7)    Line 456, The word “treatment” or “application” is missing at the end of the sentence.

8)    Line 575: It has become fashionable to postulate additional “rigorous clinical trials” in all but every paper discussing medical data. Is that sentence is maintained, it should be mentioned how unrealistic such a call is, given the low patient number and the large diversity even of those few cases.

9)    The meaning of “but” in line 587 is not clear.

Author Response

REVIEWER 2

In their manuscript, „A Literature Review Of Oral Vertical Dyskinesia (“Rabbit” 2 Syndrome) Rissardo et al. provide a systematic literature review on that highly rare disease. The last previous review on the topic is quite dated; thus the current manuscript is timely. It is also overall well written. The tables and the two figures are helpful also.

Major concern

1)    The pathophysiology and the possibly involved pathways with the many different cases mentioned are overwhelming and it is easy to lose sight of the overall picture. A Figure showing the various involved neurotransmitter pathways and the drugs interfering with those would thus be helpful (probably involving cortico-striatal, mesencephalic, basal ganglia pathways). Such a graph might possibly also be suitable as a graphical abstract of the manuscript.

Authors: Reviewer, we agree with your point of discussion, and at the first glance that we started writing the manuscript this was the idea of providing a figure with a pathophysiological mechanism. But, there is a problem with this idea because there is no possible correlated mechanism or basic studies performed with RS. A figure describing the pathophysiological mechanism will not localize to anywhere. Therefore, after we assess the literature, we thought in describing the mechanisms already proposed or try to compilate them in three sections that are “4.1. Cases Related To Relative Hypercholinergic And Hypodopaminergic State,” “4.2. Cases Unrelated To A Direct Hypercholinergic And Hypodopaminergic State,” and “4.3. Rabbit Syndrome, Drug-Induced Parkinsonism, And Tardive Dyskinesia.” At this point, the literature does not have enough information for us to presuppose a mechanism based in facts or real findings. We hope that the Reviewer would understand our point of discussion. We will provide a general description for a graphical abstract.

Minor concerns

1)    It should be better explained how the criterion “insufficient data” e.g. in Fig. 1 as a reason for the exclusion of studies was defined.

Authors: For answer another Reviewer, we removed the previous Figure, and adapted according to PRIRSMA guidelines.

2)    The quotation marks should be moved from line 60 to line 59

Authors: The quotation marks were removed. Thank you!

3)    Line 103-108 and 561-563 are a concession to the zeitgeist. It should also be mentioned that the disadvantages of such politically correct, but less descriptive altered terminology makes it harder to find relevant papers in literature databases for future scientists, as thus they would have to know all the different proposed possible designations of the disease in their literature search. Moreover, “rabbit syndrome” is much more intuitive than the proposed new wording even for the lay public.

Authors: Excellent comment by the Reviewer, we included his/her phrase in the manuscript, and we believe that significant improve its quality.

“Noteworthy, the disadvantages of such politically correct, but less descriptive altered terminology makes it harder to find relevant papers in literature databases for future scientists, as thus they would have to know all the different proposed possible designations of the disease in their literature search.”

4)    The authors mention several times that RS may be misdiagnosed as TD. The authors should check whether there are examples for such misdiagnosis in the literature, in addition to the 1979 paper by Simpson et al.

Authors: Dear Reviewer, to avoid the misdiagnosed between RS and TD, and followed the definition by Dr. Villeneuve (ref 1). Also, the reader can find in the supplementary material the description of all the cases from the literature with two interesting features that are the PD and dyskinetic features associated with RS. We reviewed every study and video published searching for these features in the patients reported. And we interestingly found the idea summarized by Figure 2, which is the overlap between syndromes that could be the missing link between these different movement disorders that are two different words of hypokinetic and hyperkinetic movements.

5)    The authors should also mention any data or experience with own patients how RS is being perceived by the patients, i.e. how bothersome it is usually perceived. For instance, such a description might fit in after lines 357-370. Such a description could be followed by an evaluation whether the side effects of e.g. anti-cholinergic treatment (e.g. on cognition), etc. are commensurate with the potential benefits.

Authors: this would be magnificent Reviewer, but the main author of the manuscript (JPR) personally never saw a patient with this syndrome. Only the senior authors (SNC) already saw some patients in his psychiatric clinic, but it was some years ago when he was still responsible for the psychiatric hospital in the region. Based in these facts, we only included the main findings from the literature that is still quite interesting to see the perception of the patients about their movement disorder.

6)    Line 258: Consider replacing “To increase the confusion” with “Increasing confusion,…”

Authors: we agree with the Reviewer, and we replaced the phrases.

7)    Line 456, The word “treatment” or “application” is missing at the end of the sentence.

Authors: we agree with the Reviewer, and we included the word therapy for fulfil this concern.

8)    Line 575: It has become fashionable to postulate additional “rigorous clinical trials” in all but every paper discussing medical data. Is that sentence is maintained, it should be mentioned how unrealistic such a call is, given the low patient number and the large diversity even of those few cases.

Authors: we thank the Reviewer for this interesting comment, and we included the following phrase to address this comment.

“It is worth mentioning that clinical trials are likely unfeasible with RS due to low patient number and the large diversity throughout the cases.”

9)    The meaning of “but” in line 587 is not clear.

Authors: We modify the “but” to the preposition “with,” for further understanding of the conclusion.

We would like to do a further comment to the Reviewer. We appreciate the time and expertise of the Reviewer; the modifications propose were significant for the improvement of the manuscript quality.

Reviewer 3 Report

Comments and Suggestions for Authors

Current review by Jamir Rissardo et al summarizes well literature data on diagnosis, pathological mechanisms and association with medication of "rabbit" syndrome. The manuscript is helpful, especially for medical doctors treating psychiatric patients.

1. Before publication, I recommend one more figure to be included summarizing all patient risk factors to develop "rabbit" syndrome;

2. How authors would explain the fact that Asian individuals were reported to be more vulnerable to drug-induced parkinsonism ?

3. More literature data should be introduced to support the interesting hypothesis of "missing link".

In general, the review is well written, informative, and I recommend its publication  after making these changes.

Author Response

REVIEWER 3

Current review by Jamir Rissardo et al summarizes well literature data on diagnosis, pathological mechanisms and association with medication of "rabbit" syndrome. The manuscript is helpful, especially for medical doctors treating psychiatric patients.

  1. Before publication, I recommend one more figure to be included summarizing all patient risk factors to develop "rabbit" syndrome;

Authors: Thank you Reviewer for this comment, we will do a graphical abstract summarizing the manuscript including risk factors and possible pathophysiology.

  1. How authors would explain the fact that Asian individuals were reported to be more vulnerable to drug-induced parkinsonism?

Authors: We would like to mention to the Reviewer that this was not the focus of the current manuscript. But, the main author of the manuscript is an expert in drug-induced movement disorders. Based on his thoughts, it is possible that there are some genetic factors involved with the Asiatic population. Based on previous own data (JPR), he observed that Asian usually manifest drug-induced movement disorders in an younger age and also in lower doses than other populations. Pharmacogenetic studies are needed to specifically define this, but it is possible that in the future we see different prescription for specific populations specially regarding the formulation of the medications and its dosages.

  1. More literature data should be introduced to support the interesting hypothesis of "missing link".

Authors: thank you Reviewer for this comment, we included the following phrase to address this comment “Currently, RS, DIP, and TD are considered different disorders, but based in our findings several individuals had features of RS associated with DIP or TD, and some individuals developed the three disorders together.” Also, it is worth mentioning that we are proposing this theory, so there are no reports in the literature for us to clearly reference as a constructive analysis.

In general, the review is well written, informative, and I recommend its publication  after making these changes.

Round 2

Reviewer 1 Report

Comments and Suggestions for Authors

The authoors corrected article according to suggestions.

I recommend to accept the paper,